# Technical efficiency evaluation of colorectal cancer care for older patients in Dutch hospitals

**Thea C. Heil**[1]*, **René J. F. Melis**[1], **Huub A. A. M. Maas**[2], **Barbara C. van Munster**[3], **Marcel G. M. Olde Rikkert**[1], **Johannes H. W. de Wilt**[4], **Eddy M. M. Adang**[5], on behalf of the PRECOLO consortium[¶]

1 Department of Geriatric Medicine, Radboud University Medical Center, Nijmegen, The Netherlands,
2 Department of Geriatric Medicine, Elisabeth-Tweesteden Hospital, Tilburg, The Netherlands,
3 Department of Internal Medicine, University Medical Center Groningen, University of Groningen, Groningen, The Netherlands, 4 Department of Surgery, Radboud University Medical Center, Nijmegen, The Netherlands, 5 Department for Health Evidence, Radboud University Medical Center, Nijmegen, The Netherlands

¶ Membership of the PreColo consortium is provided in the Acknowledgments
* Thea.Zonneveld-Heil@radboudumc.nl

**Data Availability Statement:** We have included the answers to the questionnaires and more detailed hospital characteristics, including surgical procedures, in S2–S4 Tables. The publicly available

## Abstract

### Background

Preoperative colorectal cancer care pathways for older patients show considerable practice variation between Dutch hospitals due to differences in interpretation and implementation of guideline-based recommendations. This study aims to report this practice variation in preoperative care between Dutch hospitals in terms of technical efficiency and identifying associated factors.

### Methods

Data on preoperative involvement of geriatricians, physical therapists and dieticians and the clinicians' judgement on prehabilitation implementation were collected using quality indicators and questionnaires among colorectal cancer surgeons and specialized nurses. These data were combined with registry-based data on postoperative outcomes obtained from the Dutch Surgical Colorectal Audit for patients aged ≥75 years. A two-stage data envelopment analysis (DEA) approach was used to calculate bias-corrected DEA technical efficiency scores, reflecting the extent to which a hospital invests in multidisciplinary preoperative care (input) in relation to postoperative outcomes (output). In the second stage, hospital care characteristics were used in a bootstrap truncated regression to explain variations in measured efficiency scores.

### Results

Data of 25 Dutch hospitals were analyzed. There was relevant practice variation in bias-corrected technical efficiency scores (ranging from 0.416 to 0.968) regarding preoperative colorectal cancer surgery. The average efficiency score of hospitals was significantly different

quality indicators published by the Dutch Health and Youth Care Inspectorate are available from the website of the Dutch Health and Youth Care Inspectorate: https://www.dhd.nl/producten-diensten/omniq/Paginas/Databestanden-Basisset-MSZ.aspx (Basisset MSZ 2017 and Basisset MSZ 2018). Because this data is publicly available and contains identifying information, it is not possible to link this data directly to our data (website is added to S1 Table). Data on postoperative complications are obtained from the Dutch Surgical Colorectal Audit and are limited available on request against payment: https://dica.nl/dcra/onderzoek (website is added to S1 Table). The authors had no special access privileges to data from the Dutch Surgical Colorectal that others would not have.

**Funding:** This study was supported by a grant from ZonMw (80-85009-98-1003), award to MOR. The funders had no role in study design, data collection and analysis, decision to publish, or preparation of the manuscript.

**Competing interests:** The authors have declared that no competing interests exist.

from the efficient frontier (p = <0.001). After case-mix correction, higher technical efficiency was associated with larger practice size (p = <0.001), surgery performed in a general hospital versus a university hospital (p = <0.001) and implementation of prehabilitation (p = <0.001).

## Conclusion

This study showed considerable variation in technical efficiency of preoperative colorectal cancer care for older patients as provided by Dutch hospitals. In addition to higher technical efficiency in high-volume hospitals and general hospitals, offering a care pathway that includes prehabilitation was positively related to technical efficiency of hospitals offering colorectal cancer care.

## Introduction

With nearly 60% of patients aged above 70 years and more than 35% of patients aged 75 years and over, colorectal cancer is predominantly a disease of older adults [1]. As multimorbidity is commonly prevalent from the age of 70 years and multimorbidity is associated with more postoperative complications, older patients are more prone to postoperative complications and mortality [1–3].

Regardless of age or number of co-morbid conditions, surgery is the cornerstone for curative treatment in patients with colorectal cancer [4]. To maximize treatment outcomes, it is therefore important to optimize resilience in this population to withstand colorectal surgery as a stressful event. Recognition of risk factors by a multidisciplinary team approach could help identify patients at high-risk for developing postoperative complications [5]. Especially in frail older patients a comprehensive geriatric assessment (CGA), a multidimension evaluation to identify medical, psychosocial, and functional limitations of a frail older patient, is helpful to make a tailored treatment plan taken into account goals and wishes of individual patients [6].

After preoperative risk stratification, employing interventions to prepare a patient for surgery (also called "prehabilitation") including physical therapy and nutritional assessment, could counteracting the complication risks by enhancing resilience and functional capacity [7]. Prehabilitation can be unimodal, focusing solely on for instance exercise, or multimodal including physical exercise, nutrition assessment as well as psychological stress reduction [8]. Prehabilitation has shown promising results, especially for patients at greatest risk of poor postoperative outcomes [9, 10]. However, other studies did not show significant improvement of postoperative outcomes after prehabilitation in colorectal cancer surgery [11, 12].

Currently, Dutch guidelines recommend screening on frailty and geriatric assessment in case of frailty in colorectal cancer patients aged ≥ 70 years. However, quality indicators of the Dutch Health and Youth Care Inspectorate (part of the Ministry of Health, Welfare and Sport) show that screening on frailty and geriatric assessment in case of frailty is not yet completely implemented in all Dutch hospitals [13]. On the other hand, despite level II evidence implied that multimodal prehabilitation improves postoperative outcomes [14], prehabilitation is still restricted to research settings in the Netherlands because data supporting efficiency is contradictory [15, 16]. Meanwhile, several forms of prehabilitation programs have been started in a number of Dutch hospitals [17]. Therefore, colorectal care pathways show considerable practice variation between Dutch hospitals.

This study will detail this practice variation, in terms of technical efficiency, in preoperative colorectal cancer care for older patients in Dutch hospitals. Technical efficiency is defined as the extent to which a hospital invests in multidisciplinary preoperative care in relation to its outputs in terms of postoperative complications. Further, this study will identify factors associated with the variation in the technical efficiency of this preoperative care.

## Materials and methods

### Data collection

This observational study was based on retrospective data of perioperative care given to patients with colorectal cancer of 75 years and above in Dutch hospitals. Hospital data for the year 2017 and 2018 were used. In total, 56 Dutch hospitals with an active practice of colorectal cancer surgery (about 70% of all hospitals), were approached to participate in this study.

This study combined three data sources (S1 Table) to investigate the variation in technical efficiency and its potential drivers: 1. Data on preoperative involvement of physical therapists and dieticians were collected using questionnaires (S1 File) among colorectal cancer surgeons and specialized nurses. They were asked if physical therapists and dieticians were involved in the preoperative period (yes/ no/ by indication). Additionally, clinicians' judgement on prehabilitation implementation (yes/ no/ by indication) was asked. 2. Data on preoperative involvement of geriatricians were collected using the quality indicators published by the Health and Youth Care Inspectorate. These quality indicators represent, on hospital level, the percentage of frail older patients ($\geq$70 years) which is assessed by a geriatrician [13]. It was assumed that this percentage was a representative reflection of geriatric involvement in patients aged 75 years or over. 3. Data on postoperative complications obtained from the Dutch Surgical Colorectal Audit [18]. Only postoperative data of patients with elective surgery were included, as multidisciplinary preoperative interventions are not applicable in case of urgency or emergency surgery. Patients with hyperthermic intraperitoneal chemotherapy (HIPEC) or intraoperative radiation therapy (IORT) were excluded for this analysis.

The multiple datasets were merged to make a dataset that registered all hospital-related health services delivered to these patients.

The institutional review board (IRB) CMO Region Arnhem-Nijmegen, NL55712.091.16 (file number 2018–4163) advised that this study doesn't fall within the remit of the Medical Research Involving Human Subjects Act (WMO).

### Outcomes

The primary outcome variables of the study were the scores of technical efficiency for each individual hospital and the comparisons between this scores. The technical efficiency score was defined as the extent to which a hospital invests in multidisciplinary preoperative care (input) in relation to its outputs in terms of postoperative complications.

The secondary outcome variable was the relationship between hospital technical efficiency and quality performance and the factors affecting this relationship.

### Statistical analysis

The analysis on this collapsed dataset consisted of a two-stage data envelopment approach (DEA). DEA is a non-parametric technique based on linear programming that allows for the construction of the most efficient production frontier based on the inputs and outputs of the decision-making units (DMUs: these are the hospitals delivering colorectal cancer surgery care). In other words, this technical efficiency frontier reflects the graphical line that can be

constructed when connecting the DMUs that use the least amount of inputs to produce one unit of output (input-oriented DEA) or that produces the most amount of outputs with one unit of input (output-oriented DEA).

The relative technical (in)efficiency, the difference between the DEA score and the efficient frontier is calculated in the first stage of the DEA by comparing its inputs and outputs for each DMU in relation to the rest of the DMUs, i.e. hospitals. In this study input was defined as the average costs of geriatrician, physical therapist and dietician involvement with a patient who is scheduled for colorectal cancer surgery in a period between setting operation indication and admission to the hospital because of tumor resection. Output was defined as the percentages of severe complications in each hospital. A severe complication was defined as a complication within 90 days after resection with serious consequences: leading to mortality, a surgical reintervention (operative or percutaneous), a postoperative hospital stay of at least 14 days or readmission. As lower values of severe complications represent better quality of care, and DEA usually assumes that more outputs contribute to higher technical efficiency, the percentages of no severe complications were used in the DEA analysis.

To explain differences in technical efficiency scores, the second stage of the DEA comprised a bootstrapped truncated regression analysis where estimated technical efficiency scores were regressed on a set of preselected case-mix adjusting and explanatory variables. Preselected case-mix variables were ASA score, tumor stage and tumor localization. Selected explanatory (independent) variables were hospital volume, hospital teaching status and clinicians' judgement on the implementation of prehabilitation. The complete approach is described in more detail in S2 File.

Data on population and hospital level are presented as mean (standard deviation [SD]) or number (%) when indicated. Comparisons between means and between DEA scores were done using Student's t-test for numerical variables. Comparisons between categorical variables were done using Chi-Square test. A P-value of <0.05 was considered statistically significant.

The entire analysis, i.e. Simar & Wilson approach, was carried out using STATA version 15.1.

## Results

In total 56 of the 79 hospitals (71%) conducting colorectal cancer surgery were approached. 25 out of this 56 hospitals (45%) provided sufficient information (meaning at least one questionnaire completed by surgeon or specialized nurse and available data on postoperative complications) and were taken into analysis (S2–S4 Tables). In these 25 hospitals a total of 2470 elective colorectal cancer patients of 75 years and older underwent surgery. These patients comprised 39% of the total group of patients (n = 6349) who were treated in the Netherlands during this period.Mean age of patients in the participating hospitals ranged from 78 to 81 years (Table 1).

The aggregated descriptive data on hospital level used to calculate uncorrected and bias-corrected (bootstrapped) DEA scores is shown in Table 2.

The classical DEA analysis showed that 7 hospitals (28%) were lying on the technical efficiency frontier (uncorrected DEA score = 1) and served therefore as benchmarks for the other hospitals (Table 2). The large distribution of technical efficiency scores (0.443–1), based on both variation in postoperative outcomes (output) as well as preoperative involvement of physical therapists, dieticians and geriatricians (input), is depicted in Fig 1.

Mean bias-corrected DEA score from the bootstrapped DEA analysis was 0.798 (range 0.416 to 0.968), and differed significantly from the technical efficiency frontier (p = <0.001, 95%CI -0.257, -0.147).

**Table 1. Descriptive data on population level.**

| | Patients (n = 2470) from hospitals included (n = 25) | | | | Patients (n = 3879) from hospitals not included (n = 31) | Total (n = 6349) |
|---|---|---|---|---|---|---|
| | *University hospitals (n = 148)* | *General hospitals (n = 2322)* | *Total (n = 2470)* | *P-value* | | |
| **Hospital size**[a] | 30 (10) | 116 (42) | 99 (52) | <0.001 | 75 (42) | 82 (46) |
| **Age** | 79 (4) | 80 (4) | 80(4) | 0.001 | 80 (4) | 80 (4) |
| **Sex** | | | | 0.034 | | |
| Male | 91 (61%) | 1219 (52%) | 1310 (53%) | | 2073 (53%) | 3383 (53%) |
| Female | 57 (39%) | 1103 (48%) | 1160 (47%) | | 1806 (47%) | 2966 (47%) |
| **ASA ≥ 3** | 58 (39%) | 1040 (45%) | 1098 (44%) | 0.184 | 1583 (41%) | 2681 (42%) |
| **Cancer type** | | | | 0.011 | | |
| Colon | 95 (64%) | 1712 (74%) | 1807 (73%) | | 2895 (75%) | 4702 (74%) |
| Rectum | 53 (36%) | 610 (26%) | 663 (27%) | | 984 (25%) | 1647 (26%) |
| **Tumor stage IV** | 21 (14%) | 120 (5%) | 141 (6%) | <0.001 | 186 (5%) | 327 (5%) |
| **Severe complications** | 41 (28%) | 552 (24%) | 593 (24%) | 0.278 | 952 (25%) | 1545 (24%) |

Data are n (%) or mean (SD).

[a] Average number of treated patients in each hospital in 2017–2018.

Table 3 shows the results of the truncated regression analysis. After case-mix correction the higher bias-corrected technical efficiency was significantly associated with larger practice size (β = 0.003, p = <0.001, 95%CI 0.002,0.003) and surgery performed in a general hospital versus a university hospital (β = 0.402, p = <0.001, 95%CI 0.234,0.557). Additionally, clinicians' judgement on the implementation level of prehabilitation was positively associated with hospital preoperative colorectal cancer pathway technical efficiency. Implementation of prehabilitation by indication or as usual care both showed better technical efficiency than no prehabilitation at all (respectively, β = 0.209, p = <0.001, 95%CI 0.127,0.292 and β = 0.187, p = <0.001, 95%CI 0.094,0.282).

## Discussion

This study studied technical efficiency of preoperative colorectal cancer care, based on preoperative involvement of physical therapists, dieticians and geriatricians as input, and severe postoperative complications as output. Across Dutch hospitals, considerable technical inefficiencies were shown in this study, as only 7 of the 25 hospitals were on the efficient frontier. In regression analysis, practice size, being a general hospital and implementation of prehabilitation (both by indication or as usual care) were significantly positively associated with technical efficiency.

As far as we know, this is the first study that analyzed the influence of both hospital volume and hospital teaching status on technical efficiency scores in preoperative colorectal cancer care for older patients. Previous studies already described the association between hospital volume and postoperative outcomes in colorectal cancer care as such, with reduced postoperative mortality in high volume hospitals [19–21]. Kolfschoten et al. found that this association between hospital volume and postoperative outcomes was mostly explained by high-volume hospitals treating patients with less comorbid diseases, lower ASA-classification and less often in an urgent or acute setting [22]. In this study only elective surgery was included and this study corrected for ASA-classification, therefore it is likely that other factors play a role in the variance in technical efficiency. An explanation could be that high-volume hospitals use more

Table 2. Hospital characteristics representing input and output for DEA score.

| Hospital number | Hospital size (n) | University hospital[a] | Prehabilitation[b] | ASA ≥ 3 (%) | Colon cancer (%) | Tumor stage IV (%) | Psychical therapist[b] | Dietician[b] | Geriatrician[c] | Severe complications (%) | Uncorrected DEA score[d] | Bias-corrected DEA score[e] |
|---|---|---|---|---|---|---|---|---|---|---|---|---|
| 1 | 14 | Y | I | 4 (29%) | 8 (57%) | 2 (14%) | I | I | 1 | 4 (29%) | .962 | .904 |
| 2 | 27 | Y | I | 15 (56%) | 15 (56%) | 6 (22%) | I | I | 2 | 8 (30%) | .822 | .789 |
| 3 | 31 | Y | N | 14 (45%) | 17 (55%) | 7 (23%) | I | I | 3 | 12 (39%) | .678 | .623 |
| 4 | 35 | Y | N | 10 (29%) | 21 (60%) | 3 (9%) | I | I | 3 | 8 (23%) | .852 | .784 |
| 5 | 41 | Y | N | 15 (37%) | 29 (71%) | 3 (7%) | I | I | 3 | 9 (22%) | .842 | .774 |
| 6 | 41 | N | Y | 24 (59%) | 30 (73%) | 2 (5%) | Y | Y | 2 | 8 (20%) | .757 | .700 |
| 7 | 46 | N | N | 7 (15%) | 32 (70%) | 1 (2%) | Y | Y | 3 | 15 (33%) | .443 | .416 |
| 8 | 64 | N | N | 29 (45%) | 47 (73%) | 5 (8%) | Y | Y | 2 | 26 (41%) | .590 | .548 |
| 9 | 70 | N | Y | 49 (70%) | 46 (66%) | 6 (9%) | Y | N | 1 | 18 (26%) | 1 | .768 |
| 10 | 88 | N | I | 33 (38%) | 54 (61%) | 4 (5%) | I | I | 1 | 24 (27%) | 1 | .914 |
| 11 | 98 | N | I | 35 (36%) | 74 (76%) | 11 (11%) | I | I | 3 | 20 (20%) | .859 | .775 |
| 12 | 99 | N | Y | 39 (39%) | 70 (71%) | 4 (4%) | N | Y | 2 | 20 (20%) | 1 | .923 |
| 13 | 107 | N | Y | 56 (52%) | 81 (76%) | 2 (2%) | Y | I | 1 | 22 (21%) | 1 | .901 |
| 14 | 112 | N | I | 49 (44%) | 90 (80%) | 2 (2%) | I | I | 2 | 20 (18%) | .984 | .954 |
| 15 | 116 | N | I | 59 (51%) | 83 (72%) | 8 (7%) | I | Y | 2 | 28 (24%) | .841 | .774 |
| 16 | 121 | N | I | 56 (46%) | 78 (65%) | 9 (7%) | Y | I | 3 | 23 (19%) | .798 | .719 |
| 17 | 124 | N | N | 56 (45%) | 86 (70%) | 4 (3%) | N | I | 3 | 41 (33%) | .843 | .778 |
| 18 | 127 | N | Y | 56 (44%) | 91 (72%) | 5 (4%) | N | I | 2 | 36 (28%) | .902 | .829 |
| 19 | 135 | N | Y | 54 (40%) | 103 (76%) | 7 (5%) | I | I | 1 | 34 (25%) | .988 | .945 |
| 20 | 136 | N | N | 61 (45%) | 99 (73%) | 5 (4%) | N | I | 1 | 28 (21%) | 1 | .880 |
| 21 | 140 | N | N | 77 (55%) | 101 (72%) | 2 (1%) | N | N | 3 | 37 (26%) | 1 | .717 |
| 22 | 145 | N | Y | 74 (51%) | 94 (65%) | 8 (6%) | Y | Y | 1 | 42 (29%) | .857 | .810 |
| 23 | 163 | N | I | 45 (28%) | 105 (64%) | 16 (10%) | I | I | 2 | 26 (16%) | 1 | .968 |
| 24 | 187 | N | N | 50 (27%) | 131 (70%) | 8 (4%) | I | I | 3 | 40 (21%) | .848 | .785 |
| 25 | 203 | N | I | 121 (60%) | 128 (63%) | 8 (4%) | I | I | 1 | 44 (22%) | .988 | .963 |

[a] N = no, Y = yes.

[b] N = no, I = by indication, Y = yes.

[c] 1 = involvement in < 15% of patients, 2 = involvement in 15–25% of patients, 3 = involvement in >25% of patients.

[d] Classical non-parametric DEA analysis.

[e] Bootstrapped (semi-parametric) DEA analysis.

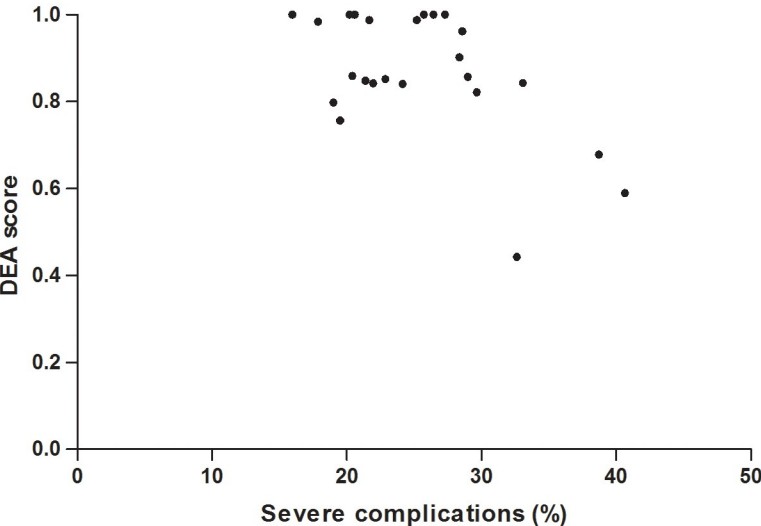

**Fig 1. Distribution of hospital uncorrected (non-parametric) DEA scores in proportion to the percentage of severe complications (output).**

frequently selection criteria to identify only those patients that require multidisciplinary pre-operative care. This hypothesis is supported by the fact that the hospital with the lowest bias-corrected DEA score was a relatively low-volume general hospital, with involvement of both physical therapist and dietician on a regular basis and involvement of a geriatrician in a relatively high number of patients. In contrast, in the hospitals with the highest technical efficiency both physical therapist and dietician were only involved by indication and the percentage of patients in which a geriatrician was involved was nearly a third in comparison with the hospital with the lowest bias-corrected DEA score.

The association between hospital teaching status and serious complication rates, with higher serious complication rates in university hospitals compared to general hospitals, was previously described by van Groningen et al. [23]. They also found that hospital volume could not explain these differences and that there was a considerable hospital variation. This hospital variation is also seen in the present study, with one of the five university hospitals represented in the top ten of most technical efficient hospitals, while another of the university hospitals had nearly the lowest bias-corrected DEA score. As this study did only correct for ASA-score and

**Table 3. Results of truncated regression analysis.**

|  | Coefficient (β)[a] | P-value | 95% confidence interval |
|---|---|---|---|
| **Tumor stage IV** | -0.012 | 0.037 | -0.023,-0.002 |
| **ASA ≥ 3** | 0.000 | 0.947 | -0.003,0.003 |
| **Tumor localization, colon** | 0.004 | 0.262 | -0.003,0.012 |
| **Practice size** | 0.003 | <0.001 | 0.002,0.003 |
| **General hospital** | 0.402 | <0.001 | 0.234,0.557 |
| **Prehabilitation (reference no prehabilitation)** |  |  |  |
| *By indication* | 0.209 | <0.001 | 0.127,0.292 |
| *Yes* | 0.187 | <0.001 | 0.094,0.282 |

[a] Positive coefficient represents increase in efficiency score.

tumor stage, it is possible that the association between technical efficiency and hospital teaching status is based on differences in case mix such as location of the tumor, type of resection or gender [24]. It is likely that more complex operations, with higher postoperative complications rates, will take place in university care settings, which might also biased the results [22].

The newly found association between technical efficiency and the clinicians' judgement on the implementation of prehabilitation may be explained by the beneficial effects of prehabilitation on postoperative outcomes as shown in the literature [25]. Additionally, it is likely that implementation of prehabilitation in a care pathway ensures that preoperative care is more structured, with probably better selection criteria of patients that need multidisciplinary preoperative care. Though, the association between clinicians' judgement on prehabilitation can also be reversed; whereby hospitals with a higher technical efficiency are more aware of (the importance of) local care pathways and therefore gave a positive answer to the question whether or not prehabilitation was implemented.

This study is the first that investigates practice variation between Dutch hospitals in terms of technical efficiency in preoperative colorectal cancer care.

This study has several limitations. Firstly, only 25 of the 79 Dutch hospitals conducting colorectal cancer surgery were included. However, the patients treated in the included hospitals were a representative patient population. Above, quantitative data on involvement of physical therapists and dieticians was not available and this study was based on subjective clinicians' judgement. In some cases, there was discrepancy between the answers of the surgeon and the specialized nurse inside the hospital. To deal with these incongruities it was chosen to select the highest value of involvement, as it was assumed that the highest value came from consultants ordering the physical therapists or dieticians involvement. Additionally, assumptions were made about the degree of involvement of the physical therapist and dietician in the participating hospitals, because it was not possible to quantify this for each participating hospital. It was also assumed that data on preoperative involvement of geriatricians in patients aged 70 years or over, collected using the quality indicators published by the Health and Youth Care Inspectorate, was a representative reflection of geriatric involvement in patients aged 75 years or over. It is possible that for some hospitals an over- or underestimation of the real involvement was made based on this assumptions. Next, this study did not correct for all potential case-mix factors in the regression analysis. A pre-specified set of case-mix variables was chosen based on relevancy, availability and clinical consensus. However, as mentioned in previous studies, variables as gender and comorbidity levels are associated with postoperative outcomes too [22, 26]. Nevertheless, it is questionable whether or not these case-mix factors influences the clinicians' judgement on prehabilitation implementation.

In conclusion, this study showed high technical efficiency variation in colorectal cancer care for older patients between Dutch hospitals. In addition to higher technical efficiency in high-volume hospitals and general hospitals, a care pathway including prehabilitation seemed to be positively related to technical efficiency meaning that hospitals that implemented prehabilitation, by indication or as usual care, are technical efficiency benchmarks for the those hospitals that did not. Further prospective research should therefore focus on cost-effectiveness of prehabilitation and selection criteria for patients that benefit the most from prehabilitation.

## Supporting information

**S1 File. Questionnaire.**
(DOCX)

**S2 File. Detailed statistical analysis.**
(DOCX)

**S1 Table. Overview of datasources.**
(DOCX)

**S2 Table. Questionnaire responses.**
(DOCX)

**S3 Table. Detailed hospital characteristics.**
(DOCX)

**S4 Table. Detailed surgical procedures.**
(DOCX)

## Acknowledgments

PRECOLO Consortium: W.A. Bemelman, Department of Surgery Amsterdam UMC, Amsterdam, The Netherlands; L.W.J. Bogert, Department of Geriatric Medicine, Fransiscus Gasthuis en Vlietland, The Netherlands; J.W.A. Burger, Department of Surgery, Catharina Ziekenhuis, Eindhoven, The Netherlands; E.C.J. Consten, Department of Surgery, Meander Medisch Centrum, Amersfoort, The Netherlands; J.W.T. Dekker, Department of Surgery, Reinier de Graaf Gasthuis, Delft, The Netherlands; P. van Duijvendijk, Department of Surgery, Gelre Ziekenhuizen, Apeldoorn, The Netherlands; M.H. Emmelot-Vonk, Department of Geriatric Medicine, Universitair Medisch Centrum Utrecht, The Netherlands; D.J. Evers, Department of Surgery, ZGT, Almelo, The Netherlands; M.C. Faes, Department of Geriatric Medicine, Amphia Ziekenhuis, Breda, The Netherlands; A.A.W. van Geloven, Department of Surgery, Tergooi, Hilversum, The Netherlands; F.R. de Graaf, Department of Geriatric Medicine, Meander Medisch Centrum, Amersfoort, The Netherlands; W.M.U. van Grevenstein, Department of Surgery, Universitair Medisch Centrum Utrecht, The Netherlands; J. Heemskerk, Department of Surgery, Laurentius Ziekenhuis, Roermond, The Netherlands; L.M.C. Hempenius, Department of Geriatric Medicine, Medisch Centrum Leeuwarden, The Netherlands; M. Huisman-Baron, Department of Geriatric Medicine, Bernhoven, Uden, The Netherlands; A. Jacobs, Department of Geriatric Medicine, Catharina Ziekenhuis, Eindhoven, The Netherlands; F. Kamerman-Celie, Department of Geriatric Medicine, St. Anna Ziekenhuis, Geldrop, The Netherlands; C.J.P.W. Keijsers, Department of Geriatric Medicine, Jeroen Bosch Ziekenhuis, 's-Hertogenbosch, The Netherlands; H.M. Klaren, Department of Geriatric Medicine, ZGT, Almelo, The Netherlands; S.A. Koopal, Department of Surgery, Medisch Centrum Leeuwarden, The Netherlands; D. Kortbeek, Department of Geriatric Medicine, Streekziekenhuis Koningin Beatrix, The Netherlands; E.V.E. Madsen, Department of Surgery, Erasmus MC, Rotterdam, The Netherlands; L.C.F. de Nes, Department of Surgery, Maasziekenhuis Panteint, Boxtel, The Netherlands; C. Oudshoorn, Department of Geriatric Medicine, Erasmus MC, Rotterdam, The Netherlands; K.C.M.J. Peeters, Department of Surgery, Leids Universiteit Medisch Centrum, The Netherlands; M.M. Poelman, Department of Surgery, Fransiscus Gasthuis en Vlietland, The Netherlands; J.M.J. Schreinemakers, Department of Surgery, Amphia Ziekenhuis, Breda, The Netherlands; C. Sietses, Department of Surgery, Ziekenhuis Gelderse Vallei, Ede, The Netherlands; F. Slootmans, Department of Surgery, Streekziekenhuis Koningin Beatrix, The Netherlands; A.B. Smits, Department of Surgery, St. Antonius Ziekenhuis, Nieuwegein, The Netherlands; M. Sosef, Department of Surgery, Zuyderland MC, The Netherlands; N. van der Velde, Department of Internal Medicine, Section of Geriatric Medicine, Amsterdam UMC, Amsterdam, The Netherlands; E.G.G. Verdaasdonk, Department of Surgery, Jeroen Bosch Ziekenhuis, 's-Hertogenbosch, The Netherlands; D.C.M. Verheijen,

Department of Geriatric Medicine, Ziekenhuis Gelderse Vallei, Ede, The Netherlands; F.J. Vogelaar, Department of Surgery, VieCuri Medisch Centrum, The Netherlands; W. te Water, Department of Geriatric Medicine, Gelre Ziekenhuizen, Apeldoorn, The Netherlands; I.J.H. Welles, Department of Geriatric Medicine, Laurentius Ziekenhuis, Roermond, The Netherlands; B.J. van Wely, Department of Surgery, Bernhoven, Uden, The Netherlands; H.L. van Westreenen, Department of Surgery, Isala, The Netherlands; B. Wiering, Department of Surgery, Slingeland Ziekenhuis, Doetinchem, The Netherlands; S.E. Wolst, Department of Geriatric Medicine, Slingeland Ziekenhuis, Doetinchem, The Netherlands.

## Author Contributions

**Conceptualization:** René J. F. Melis, Huub A. A. M. Maas, Barbara C. van Munster, Marcel G. M. Olde Rikkert, Johannes H. W. de Wilt, Eddy M. M. Adang.

**Data curation:** Thea C. Heil, René J. F. Melis, Huub A. A. M. Maas, Barbara C. van Munster, Marcel G. M. Olde Rikkert, Johannes H. W. de Wilt, Eddy M. M. Adang.

**Formal analysis:** Thea C. Heil, Eddy M. M. Adang.

**Funding acquisition:** Huub A. A. M. Maas, Barbara C. van Munster, Marcel G. M. Olde Rikkert, Johannes H. W. de Wilt.

**Investigation:** Thea C. Heil.

**Methodology:** Thea C. Heil, Eddy M. M. Adang.

**Project administration:** Thea C. Heil, Marcel G. M. Olde Rikkert.

**Resources:** Thea C. Heil, Huub A. A. M. Maas, Barbara C. van Munster, Marcel G. M. Olde Rikkert, Johannes H. W. de Wilt.

**Software:** Eddy M. M. Adang.

**Supervision:** Marcel G. M. Olde Rikkert, Eddy M. M. Adang.

**Validation:** Eddy M. M. Adang.

**Visualization:** Thea C. Heil.

**Writing – original draft:** Thea C. Heil, René J. F. Melis, Eddy M. M. Adang.

**Writing – review & editing:** René J. F. Melis, Huub A. A. M. Maas, Barbara C. van Munster, Marcel G. M. Olde Rikkert, Johannes H. W. de Wilt, Eddy M. M. Adang.

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
