## [Decision Letter · Decision Letter 0]

23 Jun 2021

PONE-D-20-38041

Technical efficiency evaluation of colorectal cancer care for older patients in Dutch hospitals

PLOS ONE

Dear Dr. Zonneveld-Heil,

Thank you for submitting your manuscript to PLOS ONE. After careful consideration, we feel that it has merit but does not fully meet PLOS ONE’s publication criteria as it currently stands. Therefore, we invite you to submit a revised version of the manuscript that addresses the points raised during the review process.

We look forward to receiving your revised manuscript.

Kind regards,

Alaa El-Hussuna

Academic Editor

PLOS ONE

Journal Requirements:

2. Please include additional information regarding the survey or questionnaire used in the study and ensure that you have provided sufficient details that others could replicate the analyses. For instance, if you developed a questionnaire as part of this study and it is not under a copyright more restrictive than CC-BY, please include a copy, in both the original language and English, as Supporting Information."

4. We note you have included a table to which you do not refer in the text of your manuscript. Please ensure that you refer to Table 3 in your text; if accepted, production will need this reference to link the reader to the Table.

5. One of the noted authors is a group or consortium [PRECOLO consortium]. In addition to naming the author group, please list the individual authors and affiliations within this group in the acknowledgments section of your manuscript. Please also indicate clearly a lead author for this group along with a contact email address.

Additional Editor Comments (if provided):

Reviewers' comments:

Reviewer's Responses to Questions

**Comments to the Author**

1. Is the manuscript technically sound, and do the data support the conclusions?

Reviewer #1: Yes

Reviewer #2: Yes

2. Has the statistical analysis been performed appropriately and rigorously? 

Reviewer #1: Yes

Reviewer #2: Yes

3. Have the authors made all data underlying the findings in their manuscript fully available?

Reviewer #1: Yes

Reviewer #2: No

4. Is the manuscript presented in an intelligible fashion and written in standard English?

Reviewer #1: Yes

Reviewer #2: Yes

5. Review Comments to the Author

Reviewer #1: This paper looks at the impact of prehabilitation for colorectal cancer therapy on outcomes for Colorectal cancer across Dutch hospitals through the involvement of geriatricians, physiotherapy and dieticians. Administrative data from the Dutch Surgical Colorectal Audit was used. A 2-stage data envelopment analysis (DEA) approach was used. 25 hospital were included between 2017-2018. Findings were that higher technical efficiency was associated with larger practices, non-university hospitals, and the use of prehabilitation. This is an interesting and important study highlighting the need for good preoperative therapy for patients undergoing surgery for colorectal cancer. However, the study does not account for a multitude of factors that do into a successful patient outcome.

Major Issues

• Were colon and rectal cancer treatment done differently? Were rectal cancers which required neoadjuvant treatment also considered as separate. Neoadjuvant has major implications for prehabilitation as the lead time to surgery is longer. Similarly, these patients may be more deconditioned heading into their operation.

• I am still not entirely sure what prehabilitation is beyond a physiotherapy consults and a geriatric consults. It seems that consideration was not made for the type of prehabilitation but rather whether a visit had occurred. The true measure of the impact of prehabilitation is difficult to discern and as such, conclusions drawn from the study are challenging to interpret. Can you please provide an overview as to what prehabilitation includes?

• I suspect that there is a heterogeneity of surgical interventions. For example, older patients may be offered limited resections for colorectal cancer with less aggressive lymphadenectomies or even transanal excisions (TEM). Were attempts made to correct for this?

• If outcomes were complications, was any consideration given for accounting for CCI scores?

• Can you explain what the ”technical efficiency frontier” is?

Minor Issues

• Can the authors clarify which patients are offered the frailty and comprehensive geriatric assessments preoperatively. Is it age over age 70?

• In stage 1 of the methods, can you please clarify what “by indication” means and how does the division of physiotherapy become 5% vs Dietician was 40%

Reviewer #2: Thank you for inviting me to review this interesting study. The authors collected data from different sources to investigate the technical efficiency in elderly patients with colorectal cancer in Dutch hospitals. They found considerable variation in technical efficiency amongst hospitals and reported that high-volume hospitals and general hospitals offering a care pathway that includes prehabilitation had better postoperative outcome. I have several comments on the manuscript as outlines below.

Introduction

“On the other hand, as there is still a lack of data supporting efficiency, prehabilitation is generally restricted to research settings”, this is not entirely true as level II evidence implied that multimodal prehabilitation improves outcomes after surgery for abdominal cancer in general (Front. Surg., 19 March 2021 | https://doi.org/10.3389/fsurg.2021.628848).

Methods

• Please state whether the data used in the study were prospectively collected.

• Consider adding a diagram entailing three data sources used in the study.

• Regarding data source#1, how was the questionnaires sent to colorectal surgeons and nurses? How many surgeons and how many nurses were invited and how many accepted the invitation? This needs to be clarified in the results section later.

• You mention postoperative complications as the study output, however; you do not include how this outcome was defined and graded (e.g. Clavien-Dindo classification). Also, were all complications included as equal, as simple wound infection was counted equivalent to major anastomotic leak or pulmonary embolism?

• You defined technical efficiency in the introduction; this definition needs to be more clarified and moved to the methods section since this is the primary study outcome.

• Add a section detailing the primary and secondary study outcomes

• How was missing data dealt with? This needs to be addressed.

• The statistical analysis section is too long and needs to be written in a more concise manner.

Results

• Table 1 illustrates the detail of the cohort used in the study and compared University and General hospitals. It would be helpful to add the p value for each comparison to clarify whether the differences were significant.

• The coefficients of the regression analysis should be added to the text, just before the 95% confidence interval.

• Since you used a regression analysis, please report its accuracy and discriminatory ability by reporting the area under the curve.

6. PLOS authors have the option to publish the peer review history of their article (what does this mean?). If published, this will include your full peer review and any attached files.

Reviewer #1: No

Reviewer #2: **Yes: **Sameh Hany Emile

---

## [Author Response · Author response to Decision Letter 0]

18 Aug 2021

Dear Editor-in-Chief,

Enclosed please find our revised manuscript PONE-D-20-38041 “Technical efficiency evaluation of colorectal cancer care for older patients in Dutch hospitals”. We thank the reviewers for their comments, and below we describe how we have addressed each one. 

Journal Requirements:

Based on the journal requirements, we have made the following adjustments:

(1) Additional information regarding questionnaire

We have developed a survey and included S2 File, containing an overview of the questionnaire in Dutch and English. 

(2) Data avalaibility

We have included the answers to the questionnaires and more detailed hospital characteristics, including surgical procedures, in S2 - S4 Tables. 

The publicly available quality indicators published by the Dutch Health and Youth Care Inspectorate are available from the website of the Dutch Health and Youth Care Inspectorate: https://www.dhd.nl/producten-diensten/omniq/Paginas/Databestanden-Basisset-MSZ.aspx. Because this data is publicly available and contains identifying information, it is not possible to link this data directly to our data (website is added to S1 Table). 

Data on postoperative complications are obtained from the Dutch Surgical Colorectal Audit and are limited available on request against payment: https://dica.nl/dcra/onderzoek (website is added to S1 Table). 

(3) Reference Table 3

The reference has been added. 

(4) PRECOLO consortium

We added the affiliations of the individual authors. 

Lead author for this group: M.G.M. Olde Rikkert, Marcel.OldeRikkert@radboudumc.nl

Reviewer #1:

(1) Were colon and rectal cancer treatment done differently? Were rectal cancers which required neoadjuvant treatment also considered as separate. Neoadjuvant has major implications for prehabilitation as the lead time to surgery is longer. Similarly, these patients may be more deconditioned heading into their operation.

We understand the reviewer’s concern regarding the differences between treatment for colon and rectal cancer and the implications of neoadjuvant treatment for both prehabilitation and postoperative outcomes. To take into account the possible differences in colon versus rectum cancer treatment and associated postoperative outcomes, we selected tumor localisation as case-mix factor to adjust for in the bootstrapped truncated regression analysis (second stage of the analysis). As a result, independent of tumor localisation, the association between technical efficiency and the three explanatory variables (practice size, general versus academic hospital and implementation of prehabilitation) persists. 

We have not considered to correct for neoadjuvant treatment. We performed a restriction in the number of case-mix factors due to limited power to do so, because of the low number of eligible patients in some of the participating hospitals. However, we included S3 and S4 Tables with additional information on i.a. neoadjuvant treatment. 

(2) I am still not entirely sure what prehabilitation is beyond a physiotherapy consults and a geriatric consults. It seems that consideration was not made for the type of prehabilitation but rather whether a visit had occurred. The true measure of the impact of prehabilitation is difficult to discern and as such, conclusions drawn from the study are challenging to interpret. Can you please provide an overview as to what prehabilitation includes?

In this study we asked the clinicians’ judgement on prehabilitation implementation (explanatory variable second stage of the analysis) separately from the actual involvement of the physiotherapist, dietician and geriatrician (input for production function DEA, stage one). This difference becomes clear in table 1: hospitals that involve the physiotherapist/dietician/geriatrician in the preoperative phase have not always implemented prehabilitation and vice versa. 

When asking whether or not prehabilitation was applied in the hospital, no criteria of prehabilitation and what it exactly should entails were given. So it is purely the clinicians’ judgement if prehabilitation is implemented or not. 

(3) I suspect that there is a heterogeneity of surgical interventions. For example, older patients may be offered limited resections for colorectal cancer with less aggressive lymphadenectomies or even transanal excisions (TEM). Were attempts made to correct for this?

We agree that there is a heterogeneity of surgical interventions and that the surgery type is associated with postoperative complications. However, we performed a restriction in the number of case-mix factors due to the low number of eligible patients in some of the participating hospitals and have therefore chosen to only correct for tumor stage and not for resection type. 

Based on this comment, we included a S4 Table with detailed information on surgical procedures. 

(4) If outcomes were complications, was any consideration given for accounting for CCI scores?

As previously mentioned, we performed a restriction in the number of case-mix factors due to the low number of eligible patients in some of the participating hospitals. Because the Charlson Comorbidity Index is only a constellation of diseases and age without taking into consideration the severity of a disease, we have chosen to correct for ASA instead of CCI. Moreoevr, ASA classification is very common to be used pre and post-surgical procedures, which allows for comparisons with other studies. This selection is also supported by literature, showing that, although CCI had a similar predictive value for 30-day mortality and prolonged length of stay after colorectal cancer surgery, the only predictive comorbidity measure for the occurrence of post-operative surgical complications was ASA score.(1) 

In S3 Table we also included information on the CCI. 

(5) Can you explain what the ”technical efficiency frontier” is?

We have added a brief explanation of the “technical efficiency frontier” in the statistical analysis section:

DEA is a non-parametric technique based on linear programming that allows for the construction of the most efficient production frontier based on the inputs and outputs of the decision-making units (DMUs: these are the hospitals delivering colorectal cancer surgery care). In other words, this technical efficiency frontier reflects the graphical line that can be constructed when connecting the DMUs that use the least amount of inputs to produce one unit of output (input-oriented DEA) or that produces the most amount of outputs with one unit of input (output-oriented DEA). The relative technical (in)efficiency, the difference between the DEA score and the efficient frontier is calculated by comparing its inputs and outputs for each DMU in relation to the rest of the DMUs, i.e. hospitals. DEA was chosen in this study because it can deal with multiple inputs and outputs and needs no assumptions about the distribution between outputs and inputs

(6) Can the authors clarify which patients are offered the frailty and comprehensive geriatric assessments preoperatively. Is it age over age 70?

We have made the following clarification in the introduction section: 

Currently, Dutch guidelines recommend screening on frailty and geriatric assessment in case of frailty in colorectal cancer patients aged ≥70 years. 

(7) In stage 1 of the methods, can you please clarify what “by indication” means and how does the division of physiotherapy become 5% vs Dietician was 40%

We have made the following clarification in the method section: 

To indicate involvement, a surgeon and/or a specialized nurse of each participating hospital indicated involvement of physical therapists and dieticians by yes (100%), no (0%) or by indication (only in a selection of patients for whom involvement was thought to be useful). Because it was impossible to quantify the level of involvement in each individual hospital, the involvement ‘by indication’ was quantified based on clinical data of the Radboud University Medical Center (Nijmegen, the Netherlands). Between 2017-2018 the physical therapist was involved in 5% of the cases in the Radboud University Medical Center and the dietician in 40% of the cases. 

Please note: 

- Based to comment (10) of reviewer 2, we enclosed this information in S2 File.

- The use of figures of the Radboud University Medical Center for other DMU’s, is mentioned as a limitation in the discussion.

Reviewer #2:

(1) “On the other hand, as there is still a lack of data supporting efficiency, prehabilitation is generally restricted to research settings”, this is not entirely true as level II evidence implied that multimodal prehabilitation improves outcomes after surgery for abdominal cancer in general (Front. Surg., 19 March 2021 | https://doi.org/10.3389/fsurg.2021.628848).

This has been rephrased as: 

On the other hand, despite level II evidence implied that multimodal prehabilitation improves postoperative outcomes(2), prehabilitation is still restricted to research settings in the Netherlands because data supporting efficiency is contradictory. 

(2) Please state whether the data used in the study were prospectively collected.

We have made the following clarification in the method section: 

This observational study was based on retrospective data of perioperative care given to patients with colorectal cancer of 75 years and above in Dutch hospitals.

(3) Consider adding a diagram entailing three data sources used in the study.

We have taken on board this suggestion and added S1 Table. 

(4) Regarding data source#1, how was the questionnaires sent to colorectal surgeons and nurses? 

Surveys were sent by e-mail via Castors EDC.(3)

This information is also included in S1 Table. 

(5) How many surgeons and how many nurses were invited and how many accepted the invitation? This needs to be clarified in the results section later.

We agree that it is important to be transparent about this and therefore we have added this information to the result section by rephrasing the first sentence of this section:

In total 56 of the 79 hospitals (71%) conducting colorectal cancer surgery were approached. 25 out of this 56 hospitals (45%) provided sufficient information (meaning at least one questionnaire completed by surgeon or specialized nurse and available data on postoperative complications) and were taken into analysis. 

(6) You mention postoperative complications as the study output, however; you do not include how this outcome was defined and graded (e.g. Clavien-Dindo classification). Also, were all complications included as equal, as simple wound infection was counted equivalent to major anastomotic leak or pulmonary embolism?

The sentence: “A severe complication was defined as a complication within 90 days after resection with serious consequences: leading to mortality, a surgical reintervention (operative or percutaneous), a postoperative hospital stay of at least 14 days or readmission.” may have caused some confusion. Let us clarify this. The Dutch Surgical Colorectal Audit collects both information on the nature as well as the severity of postoperative complications. As mentioned by the reviewer, not all complications can be counted equivalent. To be able to calculate an overall outcome measure, independent from the nature of the complications, we used only information on the severity of complications. As described in the manuscript, only complications with serious consequences (leading to mortality, a surgical reintervention (operative or percutaneous), a postoperative hospital stay of at least 14 days) or readmission were registered. A simple wound infection was therefore not taken into analysis. 

(7) You defined technical efficiency in the introduction; this definition needs to be more clarified and moved to the methods section since this is the primary study outcome.

 As written in our response to comment (5) of reviewer 1, we have added a brief explanation of the “technical efficiency frontier” in the statistical analysis section.

(8) Add a section detailing the primary and secondary study outcomes

We added a section detailing the outcomes. 

The primary outcome variable of the study was the score of technical efficiency for each individual hospital and the comparisons between this scores. The technical efficiency score was defined as the extent to which a hospital invests in multidisciplinary preoperative care (input) in relation to its outputs in terms of postoperative complications. 

The secondary outcome variable was the relationship between hospital technical efficiency and quality performance and the factors affecting this relationship.

(9) How was missing data dealt with? This needs to be addressed.

There was only missing data for the questionnaires (S2 Table). If only the surgeon or the specialized nurse completed the questionnaire after multiple reminders by e-mail, only one questionnaire was used for analysis. This is addressed as limitation in the discussion. 

(10) The statistical analysis section is too long and needs to be written in a more concise manner

We rephrased the statistical analysis section and added S2 File for additional information on the statistical analysis. 

(11) Table 1 illustrates the detail of the cohort used in the study and compared University and General hospitals. It would be helpful to add the p value for each comparison to clarify whether the differences were significant.

We have included p-values for each comparison between University and General hospitals in table 1 and explained the statistical analysis in the method section. 

(12) The coefficients of the regression analysis should be added to the text, just before the 95% confidence interval.

We agree and have changed this in the result section.

(13) Since you used a regression analysis, please report its accuracy and discriminatory ability by reporting the area under the curve.

As this is not a study on diagnostic performance we cannot add accuracy, discriminatory ability nor AUC. We did report a secondary multivariable analysis to identify relevant drivers of efficiency, but these cannot be considered diagnostic variables.

We hope to have addressed your comments satisfactorily and thank you once more for reviewing our manuscript. 

Yours sincerely,

Thea Zonneveld-Heil, MSc

References

1. Dekker JWT, Gooiker GA, van der Geest LGM, Kolfschoten NE, Struikmans H, Putter H, et al. Use of different comorbidity scores for risk-adjustment in the evaluation of quality of colorectal cancer surgery: Does it matter? European Journal of Surgical Oncology (EJSO). 2012;38(11):1071-8.

2. Waterland JL, McCourt O, Edbrooke L, Granger CL, Ismail H, Riedel B, et al. Efficacy of Prehabilitation Including Exercise on Postoperative Outcomes Following Abdominal Cancer Surgery: A Systematic Review and Meta-Analysis. Frontiers in Surgery. 2021;8(55).

3. Castor EDC. Castor Electronic Data Capture 2019 [27 Aug. 2019]. Available from: https://castoredc.com.

---

## [Decision Letter · Decision Letter 1]

19 Nov 2021

Technical efficiency evaluation of colorectal cancer care for older patients in Dutch hospitals

PONE-D-20-38041R1

Dear Dr. Zonneveld-Hell

We’re pleased to inform you that your manuscript has been judged scientifically suitable for publication and will be formally accepted for publication once it meets all outstanding technical requirements.

Kind regards,

Alaa El-Hussuna

Academic Editor

PLOS ONE

---

## [Editor Report · Acceptance letter]

10 Dec 2021

PONE-D-20-38041R1 

Technical efficiency evaluation of colorectal cancer care for older patients in Dutch hospitals 

Dear Dr. Heil:

I'm pleased to inform you that your manuscript has been deemed suitable for publication in PLOS ONE. Congratulations! Your manuscript is now with our production department. 

Kind regards, 

on behalf of

Dr. Alaa El-Hussuna 

Academic Editor

PLOS ONE